# Impact of Covid-19 on the Mining Sector and Raw Materials Security in Selected European Countries

Andrzej Gałaś [1,*] , Alicja Kot-Niewiadomska [1] , Hubert Czerw [1] , Vladimir Simić [2] , Michael Tost [3] , Linda Wårell [4] and Slávka Gałaś [5]

1 Division of Mineral Policy, Mineral and Energy Economy Research Institute, Polish Academy of Science, Wybickiego 7A, 31-261 Krakow, Poland; a.kn@min-pan.krakow.pl (A.K.-N.); hczerw@min-pan.krakow.pl (H.C.)
2 Faculty of Mining and Geology, University of Belgrade, 11000 Belgrade, Serbia; vladimir.simic@rgf.bg.ac.rs
3 Department of Mineral Resources Engineering, Montanuniversität Leoben, 8700 Leoben, Austria; michael.tost@unileoben.ac.at
4 Economics Unit, Luleå University of Technolgy, 971 87 Luleå, Sweden; linda.warell@ltu.se
5 Faculty of Geology, Geophysics and Environmental Protection, AGH University of Science and Technology, al. Mickiewicza 30, 30-059 Krakow, Poland; sgalas@agh.edu.pl
* Correspondence: agalas@min-pan.krakow.pl; Tel.: +48-126-171-667

**Abstract:** Events that change the global economy rapidly, without warning, in principle strongly affect mining, which is one of the pillars of global development. After the first months of the Covid-19 pandemic, the mining pillar seems to be relatively stable. In this study, thanks to the meeting of an international team, it was possible to collect and compare a set of data on the impact on mining. In contrast to the general assessments of the stability of the mining sector, the authors decided to assess the impact of Covid-19 at individual stages of the mining project life cycle. In this way, it was possible to identify the most impacted fragments of the mining pillar. It was assessed that the highest influence of Covid-19 is observed in projects implementing feasibility studies and in projects for the development of new mines. The same is true of extracting residual resources in mines prior to the closure decision. The medium impact was confirmed at the exploration and discovery stage. The authors conclude that the impact on the current mining production is smaller and the effects in this case are short term, which is mainly due to a continued strong demand for minerals in China, which has balanced the weaker demand in other parts of the world. On the other hand, stopping the exploration and development of new mines will have a long-term impact, including an increased possibility of disruption of the future security of supplies of raw materials.

**Keywords:** mining project life cycle; Covid-19; raw material security; Europe

## 1. Introduction

The World Health Organization's Country Office in the People's Republic of China on 31 December 2019 found, on the website of the Wuhan Municipal Health Commission, information about a 'viral pneumonia' in Wuhan, China. On 30 January 2020, the WHO declared the novel coronavirus outbreak a Public Health Emergency of International Concern (PHEIC), which is the highest level of alarm [1]. Since then, there has been a Covid-19 pandemic in the world, with the possible exception of isolated places. In a strongly connected and integrated world, the impacts of the disease beyond mortality (those who die) and morbidity (those who are unable to work for a period of time) have become apparent since the outbreak. Amidst the slowdown of the Chinese economy with interruptions to production, the functioning of global supply chains has been disrupted [2,3]. Experience from previous pandemics shows that uncertainty levels due to pandemics are high especially in developing countries due to its strong relationship with market volatility and economic uncertainty [4].

The outbreak of the Covid-19 pandemic has disrupted the political, economic, financial and social structures all over the world. Over the past year, the impacts of the pandemic have reached far; e.g., it has limited communication, trade, access to all kinds of goods, and it almost stopped tourism and services [5]. Moreover, it lowered the demand for and sales of products and stopped many economic activities, including mining [6–8], which is the subject of our work. The current crisis can certainly be described as unprecedented. As the only one in modern history, it limited the professional and private activity of millions of people around the world. It is compared only to the Spanish flu which swept over the world after War World I [9]. Most importantly, some panic among consumers and firms has distorted usual consumption patterns and created market anomalies. Undoubtedly, it is influenced by activities to limit the spread of the virus—quarantine, travel bans and restrictions, social distance enforcement, and lockdown—closure of public places and cancellation of public events. These containment measures put in place to reduce health outcomes of the global pandemic have affected environmental sustainability and economic development [10,11]. Although global economic output is recovering from the collapse triggered by Covid-19, it will remain below pre-pandemic trends for a prolonged period [12]. According to the OECD data, global GDP decreased by 4.2% in 2020, while it is expected to increase by the same amount in 2021. Therefore, the global economy is projected to return to pre-coronavirus levels by late 2021, but the process will most likely be different across countries [13].

Covid-19 infections have been diagnosed also in all European countries [14]. The large majority of the initial prevention measures were taken in mid-March 2020. Most of the prevention measures and restrictions were kept for the whole of April. In May, several of the measures were abandoned or at least reduced in scope and severity, which had a strong effect on industrial production. This recovery continued in June and to a lesser degree in July. With increasing Covid-19 cases after the summer holidays, several countries re-introduced some containment measures in September and October. The measures were further increased in November, but in December, some countries loosened the measures during the Christmas season. Data available for the European Union show that industrial production in the EU decreased by 1.2% in December 2020 after an increase of 2.3% in November (Table 1). After strong declines in March and April 2020, industrial growth picked up in May, June, and July and remained rather stable in the subsequent months. Despite the recent decline, the total production level is now again rather close to the level before the crisis (98%) [15]. Growth rates for industries relevant to mining indicate the highest decline for crude oil and natural gas extraction (more than 20%). A much smaller decline was recorded for the following trades: mining metal ores together with other non-metallic mineral products and other mining and quarrying (below −5%). Mining of coal and lignite is at the opposite extreme, and this is the only industry with a positive result (Figure 1). It should be mentioned that most governments have allowed mining to continue during the Covid-19 pandemic, if not as per normal then with somewhat limited restrictions relating to Covid-19 mitigation [16]. This distinguishes this branch of the economy from other industries that have been closed (with short breaks) for several months due to governments decisions. Additionally, significant changes have been observed in the economic and social policy of most countries, especially in the aspect of government interventionism [17,18].

The dynamically developing pandemic affecting many spheres of social and economic life resulted in many scientific and review publications [5,7,8,10,11]. Many of them are connected with mining and the broadly understood management of mineral resources in pandemic conditions. The international consulting companies describe the response to Covid-19 and navigation of the impact on mining, including mainly metals industries [19–22]. There are science studies to predict the future impacts on metal mining [16] as well as current impact on employment in mining [23].

**Table 1.** Industrial production growth rates for total industry in EU and in the countries concerned, 2020 (%). Source [19].

| Country | April Compared to February | December Compared to April | December Compared to November |
|---|---|---|---|
| EU-27 | −26.9 | 34.1 | −1.2 |
| Euro-area | −27.5 | 34.8 | −1.6 |
| Austria | −21.7 | Not available | Not available |
| Poland | −27.0 | 40.5 | 0.5 |
| Serbia | −19.4 | 26.8 | 3.1 |
| Sweden | −15.8 | 17.7 | 0.1 |

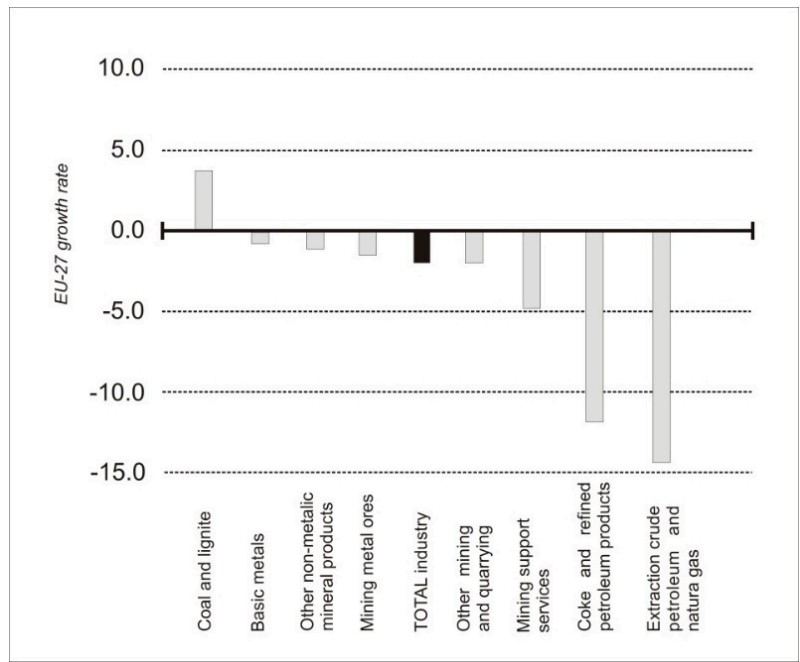

**Figure 1.** EU-27 growth rates for the mining sector and total production of all industries: December compared with February 2020 [19].

As the Covid-19 crisis is global, everyone is expecting major economic implications and problems now and directly after the outbreak of the pandemic [24]. This is the reason why impacts of Covid-19 should be considered both in the short and long term. The present work focuses on assessing and explaining the problems of the mining sector in four countries: Austria, Poland, Serbia, and Sweden (Figure 2). Authors were assessing and explaining the problems of the mining sector with special emphasis on the deposit development cycle to determine the impact of the pandemic on the security of raw material supplies, which is a novel approach to impact assessment. For the sake of simplicity, we assumed four basic stages of mining project management: (1) Exploration and discovery, (2) Feasibility and development, (3) Production, and (4) Mine closure. The article collects data in selected countries, which differ significantly in terms of available raw materials and how the industry is managed. As a result, the levels of economic risk in these countries also differ significantly in the context of raw materials management. A diverse range of aid measures may also be used in other countries. Consequently, the aim of this article is to identify potential short-term and long-term impacts of the Covid-19 pandemic on mining and mining companies in selected European countries. Impact determination was a basis for assessment of the risk of disturbance of raw materials security in designated countries.

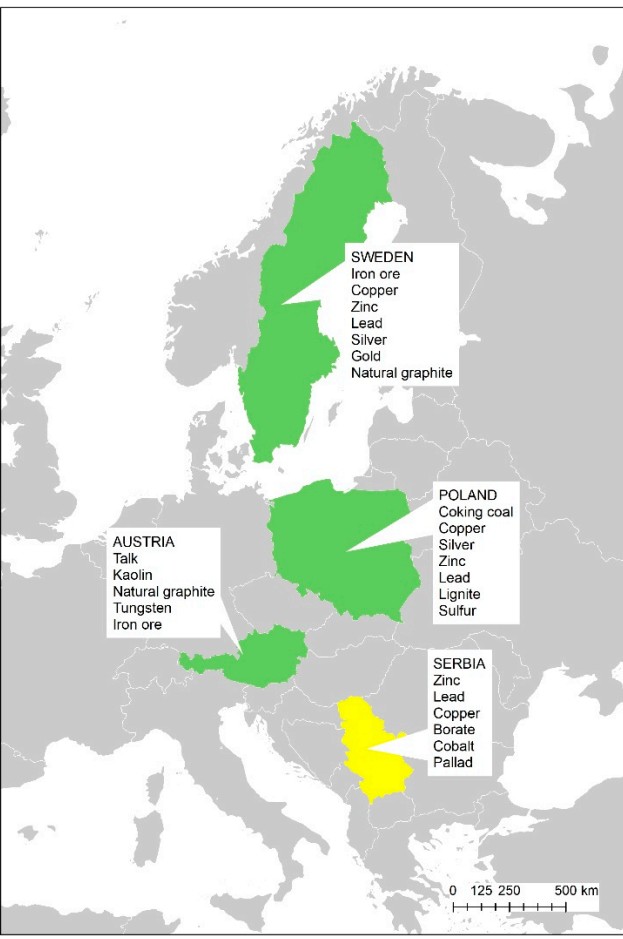

**Figure 2.** Analyzed countries (green—EU country, yellow—non-EU country) and their main resources.

The results based on the analysis of countries with different raw material potential and mining focused on different raw materials, providing a reliable assessment of the impact of Covid-19 on the mining sector. In addition to published data based on the volume of mining production, it indicates the most sensitive stages in the mining project life cycle. The article introduces important information for the forecast and management of mineral resources, in particular in the context of securing future supplies. The article also undertakes issues related to the security of supplies of selected critical raw materials, in accordance with their current list [25].

## 2. Materials and Methods

The article presents analyses and assessments of current conditions of the mining sectors in Austria, Poland, Serbia, and Sweden in order to analyze the potential effects of the Covid-19 pandemic. The basis for consideration was an appropriate questionnaire including the following elements:

- Name of Mine or Project under development,
- Type of mineral,
- Total resources,
- Production on hold or postponed,
- Losses for the country's economy,
- Losses for the EU economy,
- Time to reactivation,
- Commenced.

The questionnaire was used in each of the four countries using available data; each of the authors searched for data according to this scheme. The authors conducted surveys

in 51 mining centers or projects and with 38 mining companies. The questionnaire was used in each of the four countries using available data; each of the authors searched for data according to this scheme. For this purpose, we used direct contacts with companies and data from official company reports or government statistics as well as press releases (industry portals and websites of mining companies). Many companies provided the requested information, but it was not possible to disclose their name due to privacy reasons. The companies selected in a given country have a significant share in domestic mining production and play a key role in the international trade of raw materials. The focus was on the mining of metal ores (Fe, Cu, Zn, Pb) and selected fossil fuels (lignite, hard coal, coke) and chemical resources (sulfur).

The survey questions included a subjective overall impact ranging from (1) low—health problems of the staff and work organization; (2) medium—reduction of production, resignation from exploration programmes searching; to (3) high—existential threat. Interviews were conducted with three directors of leading mining companies.

The complement of obtained data were statistical data from databases in the four countries. Data from the Austrian Mining Handbook were used, which were collected by the Federal Ministry for Agriculture, Regions and Tourism—https://www.bmlrt.gv.at/service/publikationen/bergbau/oesterreichisches-montan-handbuch-2020.html. Data from Poland came from the MIDAS database (http://geoportal.pgi.gov.pl/midas-web) and The Balance of Mineral Resources Deposits in Poland (http://geoportal.pgi.gov.pl/surowce) prepared by the Polish Geological Survey. In Serbia, we used data from https://www.stat.gov.rs/en-US/aktuelni-pokazatelji. The Geological Survey of Sweden maintains geological (https://www.sgu.se/globalassets/mineralnaring/mineralstatistik) and mining (http://resource.sgu.se) databases that were used in this study.

The results of the analyses in the individual countries covered by the analysis were compared.

## 3. Results

The selected countries—Austria, Poland, Serbia and Sweden—differ significantly in terms of available raw materials and how the mining industry is managed. However, it is assumed that the mining project life cycle does not deviate from the standard [26,27] and can be divided into four main stages: (1) Exploration and discovery (2) Feasibility and development, (3) Production, (4) Mine closure. At each of the stages, the potential impact of the pandemic could be recognized. It may be related primarily to the availability of the workforce and the availability and changes of prices of goods and services necessary for mining activity.

### 3.1. Possible Impact of Covid-19 at Different Stages of the Mining Project

Considering the foregoing, in order to assess the impact of Covid-19 on the mining sector, its effects should be considered at a given stage of the mining project development cycle. It may be related primarily to the access to finance, changes of prices of commodities, the availability of the workforce, and services necessary for mining activity. These factors could affect mining projects in general.

Each of the above-mentioned stages of the mining project development has a certain specificity. Each standard risk stage is a period of risk related to its specificity, and in the event of a pandemic risk, the risk is additionally intensified [26,28]. In connection with the above, the article tries to identify the effect of Covid-19 on a given stage.

3.1.1. Stage 1—Exploration and Discovery (Greenfield Projects)

This stage is characterized by conducting exploration works, and in the case of encountering a potential object, detailed recognition of the deposit is undertaken. The data gathered from exploration work are used to determine whether a company will continue to explore and develop a mineral resource [29]. In general, the exploration stage requires incurring continuous costs without the possibility of generating profit. Apart from the fact

that it is cost-intensive, it is particularly burdened with the risk resulting from the lack of a guarantee of success in the form of deposit discovery or sufficiently large resources. In connection with the above, the main difficulty related to the conducting works at this stage is to obtain appropriate financing. In the face of the general economic downturn caused by Covid-19, obtaining financing may be additionally difficult. In the short term, Covid-19 should not have a negative impact on this stage, as most of the already started projects had secured funding (Table 2). However, in the long term, we can expect a decrease in the activity of junior companies, resulting from the lower availability of financial support, which in turn could lead to a reduction in the number of new deposits. The effects of these phenomena may only be felt in a few years.

**Table 2.** Potential Covid-19 impact on stages of deposit life cycle in analyzed cases.

| Stage | Potential Covid-19 Impact |
| :---: | :---: |
| Exploration and discovery | Long term |
| Feasibility and development | Short term, Long term |
| Production | Short term |
| Closure | Short term |

Proper identification of the deposit (resources, mineral quality, variability of structures) is of great importance for the decision to start a mine and its future prospects (Figure 3). Moreover, mineral exploration is the first stage in the process of mineral supply [30]. The specificity of the deposit management is that at all stages of its development, it is strongly affected by a number of conditions; e.g., the price of the raw material, and the political, social, and economic situation. All of these can change relatively easily, which makes mining investments risky, and often, the return of capital investments comes with a long delay.

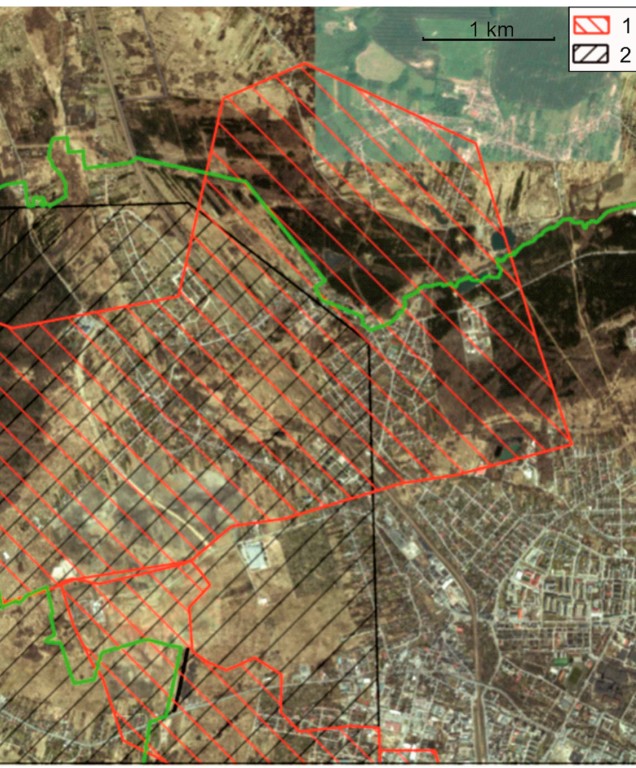

**Figure 3.** Housing estates of the city of Zawiercie built on the surface above the documented Zn-Pb deposit Zawiercie [31,32]. The delay in the decision on mining made exploitation unprofitable today. **1**—Zn-Pb documented deposit, **2**—exploration concession area.

### 3.1.2. Stage 2—Feasibility and Development (Brownfield Projects)

This stage applies to the deposits that have been already discovered, explored, and identified to the extent that it is possible to plan their development. This applies in particular to a level of geological knowledge and the amount of resources in the deposit.

At the beginning of this stage, usually, an Opportunity Study is performed, which gives an overall picture of the profitability of a possible project [33,34]. In case of positive effects, further technical and design works are carried out until the Feasibility Study is completed. Therefore, the first part of this stage is characterized by carrying out the appropriate studies without incurring such high costs as in the Prospection and exploration stage, but the last phase of this stage is crucial. It is about deciding to build a mine, which requires several times more funding than the Prospection and exploration stage. In a situation of economic uncertainty caused by Covid-19, the very decision to build a mine, even in the case of an owner with adequate resources, carries additional risk. However, in the case of the need to obtain full financing, the problem is additionally aggravated. The impact of Covid-19 in this case is both short term and long term (Table 2), because in the event of abandoning a given project, environmental and social conditions may change in the future, which might make it impossible to implement the project. Mining is a branch of industry that, apart from supplying raw materials for many other industries, is associated with harmful effects on other environmental resources, and with social and land use conflicts [35]. This is due to the fact that mining is tied to the location of the deposit, which results from the geological structure. Therefore, it is often impossible to avoid conflicts by selecting a different location. It should be noted that only after finding and documenting the resources, is it possible to propose their safeguarding, for example in land use planning instruments [36,37].

### 3.1.3. Stage 3—Production

Stage 3 is the most stable stage in the mining cycle. Obviously, the stock prices may fluctuate up and down during the production phase depending on the economic conditions, but companies can offset volatility and help maintain their value by efficient extraction and mine operation. The discovery of a new deposit in the vicinity of the operating mine may also increase mine life and stock value. If no new deposits are found, then production winds down as reserves decrease, moving the mine toward the closure stage [38]. The main investment costs have already been incurred, and profit is being generated from sales. Therefore, only the long-term persistence of prices below break-even levels is capable of shocking the activities of the mines. It is a rare situation that manages to close a mine while it is in operation and deals with project issues in declining cards. A decline caused by Covid-19 may negatively affect this stage in the event of a long-term weakening of commodities markets. The main negative impacts of Covid-19 on the production stage of mines include the reduction of production (Table 2). It can be caused by several main reasons. The first reason is the reduction in the demand for raw materials resulting from the condition of economies on a global scale. The second reason is related to pandemic limitations (workforce), the problem with transport, and market sales problems (disturbance of the product supply). Production limitations may also result from disruptions in the supply of materials necessary for mining operations.

### 3.1.4. Stage 4—Mine Closure

This stage is characterized by shutting down the mining production until the mine is completely closed, either as a result of the depletion of the deposit's resources or as a result of an unfavourable economic situation (low level of commodities prices). In the case of mines that are in the last stage of production, a sudden decline of prices and a collapse of the commodities markets with no prospects for their rapid improvement may accelerate the decision to terminate the activity (Table 2). This applies in particular to projects whose completion was planned in the near future (one, two years). Therefore, Covid-19 could accelerate the decision to close a mine that was below the break-even point.

### 3.2. Possible Impact of Covid-19 at the Mining Project in Selected European Countries

The European mining industry has a long, profitable, and varied history [39] and is fundamental for the continent's economic well-being. The consumption of aggregates, industrial minerals, and metals in Europe has grown rapidly over the past decade. Today, Europe is almost self-sufficient in producing many industrial minerals and aggregates. However, it is a significant net-importer of most metals and metal ores [40]. Mining is an important sector of development in most EU countries including Austria, Poland, Sweden, and also Serbia (outside the EU). Moreover, in the analysed countries, mining has long traditions and still has a significant impact on their economic development [41,42].

#### 3.2.1. Austria

In 2019, 1444 mine sites were in operation in Austria, producing about 80 million tons and employing over 3400 people, with the large majority being in sand and gravel production [43]. Hence, the country is self-sufficient in the field of construction raw materials. However, when it comes to metallic raw materials and industrial minerals, Austria is highly dependent on imports. In Austria, only iron ore and tungsten are currently being extracted and smelted. Plastics also need locally obtained additives such as talc, kaolin, lime, etc., but the majority of raw materials need to be imported. Austria's mines may not be the largest in the world but the country does hold a strong position in European and world markets for some minerals [44]. Globally, Austria is the number 6 producer of magnesite, and it ranks 7th for tungsten and 14th for talc [45].

Exploration projects in Austria have not encountered any particular problems with Covid-19, although delays are increasing the costs of the projects. For example, the European Lithium company, which is at the feasibility stage for a lithium (and some beryllium) deposit with estimated resources of 10.98 Mt of ore in Wolfsberg (Carinthia) [46], reported significant delays in work (Figure 4). The spodumene-bearing pegmatites have been documented over along strike, to about 450 m down dip, with average thickness of 2 m. Spodumene is the only lithium-bearing mineral [47]. Problems with settling contracts for drilling and other works resulted from travel restrictions for specialists from outside Austria. Further delays are expected in the testing of the pilot metallurgical plant in Hirschau, Germany [48]. However, another developer, Richmond Minerals INC, reported on July 27 that sample verification had been completed and field work had commenced without problems in the Oberzeiring (Styria) polymetallic (Au, Ag) deposit [49].

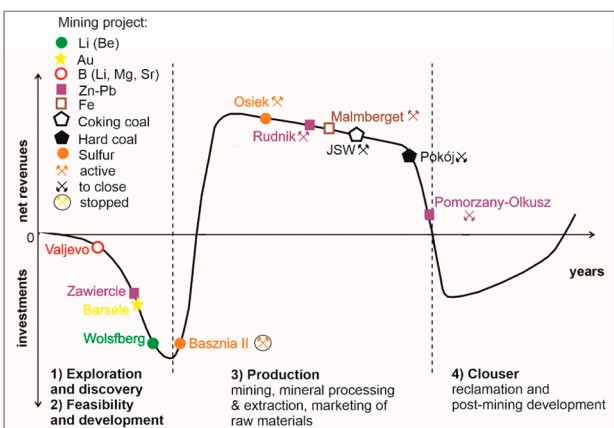

**Figure 4.** A cash flow in life cycle of a mining project. Note relatively high costs and extensive period of reclamation and post-mining development [50]. Selected mining projects in Poland discussed in the further part of the work in the current stage of development are marked.

The contribution of mining to the Austrian GDP today is minimal. However, due to the importance of the domestic metal and metalworking industry, securing the necessary imports, diversifying the sources of domestic supply, and reducing the dependency on im-

ports while maintaining market economy principles are considered particularly important. These aspects, amplified through Covid-19 related supply chain constraints, now play a key role in the current development of a national raw material strategy.

As regards Austrian mining itself, the impact on the revenues of mining companies surveyed is on average $-15\%$, ranging from $-45\%$ for graphite to zero for salt. The majority of mines also saw a reduction of production, ranging from $-40\%$ for graphite to zero for salt and tungsten, with an average production reduction of $-10\%$. One of the companies responding to the survey said that they had to close operations for a period of two weeks.

This also had an impact on employment for some of the companies: one company reduced its workforce permanently by around 20% and two others had to participate in a state-subsidized short work program, where employees worked shorter hours with the Austrian government paying a large portion of their salaries in order to prevent companies from reducing their workforce permanently.

Responding to the subjective question of the overall business impact of Covid-19, the medium hit raw materials are tungsten and graphite, with none of the Austrian survey participants experiencing any business upside.

### 3.2.2. Poland

According to data as of 31 December 2018, in total, 14,532 mineral deposits have been identified in Poland. Amongst them, more than 5000 are exploited. Altogether, over 500 million ton of minerals are mined in Poland, approximately 68% of which are rock minerals (mainly sands and gravels as well as construction and road stones), 24% are energy minerals (mostly bituminous coal and lignite), and 6.5% are metal ores, mainly Cu-Ag [51]. For many years, Poland has been one of European leaders in extraction of copper ores, silver, bituminous coal (including coking coal), and lignite [45]. The extraction industry in Poland generates about 5.4% of Gross Domestic Product. Mining ranks fourth among the industries that generate GDP. Among the largest mining companies are KGHM Polska Miedź S.A. (copper, silver), Polska Grupa Górnicza (hard coal), Jastrzębska Spółka Węglowa S.A. (coking coal), and Polska Grupa Energetyczna (lignite). It should be mentioned that coking coal extracted in Poland is one of the critical raw materials for European Union [25].

In April 2020, the owner of the Olkusz-Pomorzany zinc and lead mine announced a decision on ending mining at the end of the year. Originally, it was planned to close mining operations at the end of 2021, but the radical changes on the market in the period December 2019–April 2020 caused by the outbreak of the coronavirus pandemic accelerated the decision to end mining earlier (Figure 4). The Olkusz-Pomorzany mine has been the only operating zinc and lead mine in Poland, which has had a long tradition. For many years, over 2.6 million tons of ore were mined annually with a zinc content of over 4% and over 1.7% of lead. The extraction volume and the percentage of metals had been falling for several years. In 2019, the output was only 1.6 million tons, the zinc content was 2.6%, and that of lead was 1.3%. At the end of 2019, the reserves were at the level of 3.7 million tons with potential increments. The decision on the accelerated closure of the Olkusz-Pomorzany mine had also consequences for the possible implementation of a new mining project (Table 3). In the information dated November regarding a new contract for the feed supply, the company announced that it was withdrawing from the implementation of a new mining project using the existing underground infrastructure of the Olkusz-Pomorzany mine.

Table 3. Comparison of the effect of Covid-19 on selected mines and mining companies in the analysed countries.

| Country | Project | Type of Mineral | Stage of Life Cycle of a Mining Project | Covid-19 Impact | | | Comments | Forecast of Post-Crisis Changes |
| | | | | Production (%) | Sales (- %) | Impact on Domestic Raw Material Security | | |
|---|---|---|---|---|---|---|---|---|
| Austria | Wolfsberg | Li, Be | Feasibility | - | - | - | Significant delays | first ore scheduled for mid 2023 |
| | Erzberg | Iron ore | Production | YES | YES | NO | | uncertain of long-term impact |
| | - | Tungstein | Production | YES | YES | NO | | |
| | - | Graphite | Production | YES | YES | NO | | permanent loss to Asian competitors expected |
| Poland | Basznia II | Native sulfur | Early production | YES 100 | YES | NO | First production achieved in 2019; | possible acceleration of production |
| | Osiek | Native sulfur | Production | YES <40 | YES | NO | | possible acceleration of production |
| | JSW | Coking coal | Production | YES <3.4 | YES | NO | | possible acceleration of production |
| | Olkusz-Pomorzany | Zn-Pb | Late production—mine closure | YES 100 | NO | YES | | after securing it is impossible to reactivate the mine |

Table 3. *Cont.*

| Country | Project | Type of Mineral | Stage of Life Cycle of a Mining Project | Covid-19 Impact | | | Comments | Forecast of Post-Crisis Changes |
| --- | --- | --- | --- | --- | --- | --- | --- | --- |
| | | | | Production (%) | Sales (- %) | Impact on Domestic Raw Material Security | | |
| Serbia | Valjevo-Mionica basin | B, Li, Mg Sr, | Exploration early stage | - | - | NO | Exploration is terminated temporary due to lockdown | |
| | Čukaru Peki | Cu, Au | Early production—opening | - | NO sales | NO | Preparation for underground exploitation started in 2020, | first ore scheduled for mid 2021 |
| | Cerovo | Cu | Early production | NO | NO | NO | Preparation for exploitation started in 2019, first production achieved in 2020 | |
| | Majdanpek | Cu | Production | NO | - | NO | | |
| | Rudnik | Pb-Zn | Production | NO | YES | NO | | |
| | Grot | Pb-Zn | Production | NO | YES | NO | | |
| | Veliki Majdan | Pb-Zn | Production | NO | YES | NO | | |
| Sweden | Barsele | Au | Exploration | - | - | - | Drilling delayed for few months | |
| | Kirunavaara | Iron ore | Production | NO | NO | NO | | |
| | Malmberget | Iron ore | Production | YES | NO | NO | Affected production temporarily | |
| | Aitik | Copper | Production | NO | NO | NO | | |

The JSW Group is the largest producer of high quality hard coking coal in the European Union and one of the leading producers of coke used for smelting steel. The production and sale of coking coal and production and sale of coke and hydrocarbons constitute the JSW Group's core business. Coal extracted by the JSW Group, mainly coking coal, is used in Central Europe by local steel mills owned by international steel producers and regional utilities. The high-quality coke produced by the JSW Group is also sold on the global market. The main customers of the JSW Group's products are from Poland, Germany, Austria, the Czech Republic, Slovakia, Italy, and India. Approximately 50% of coking coal produced by the JSW Group is processed in its own coking plants. As a result, the group ultimately offers more processed and higher value products [52].

The preventive change in the organization of work and the high absence had an impact on following elements of conducted activities [53]:

- The level of coke production (−3.4% in 9 months of 2020 in comparison to 9 months of 2019);
- The level of hard coal production (−2.3% in 9 months of 2020 in comparison to 9 months of 2019);
- The total coal sales (−1.2% in 9 months of 2020 in comparison to 9 months of 2019);
- The amount of work to make resources accessible (−1.1% in comparison to 9 months of 2019);
- Timely payment of public law liabilities;
- JSW's labour costs.

As a result, the revenues on sales for quarter three of 2020 were lower (−24.5%) in the same period the year before (Table 3). It should be noted that the significant impact on the financial results of JSW was affected by price changes, as outlined below.

- Metallurgical coal price (−32.4% vs. 9 months of 2019, average price);
- Coke price (−32.7% vs. 9 months of 2019, average price).

In September 2020, the Polish government presented a list of hard coal mines to be closed. Hard coal production in Poland in 2019 amounted to 64 million tons [51]. It should be noted that coal mines reach deeper each year, where tectonic and temperature conditions are more difficult. The cost of extraction and financial risk also increase [54]. Due to its location in the most urbanized area of Poland—the Upper Silesian Agglomeration—environmental impacts constitute a serious problem in the development of cities on the surface [55]. In 2021, the Pokój and Brzeszcze mines are to be closed (Figure 4). The main reasons include the poor economic performance of these mines, which is caused by the high costs of extracting residual resources.

Significant negative impact was also seen in Polish sulfur mining. It should be mentioned that Poland is the only global producer and exporter of elemental sulfur produced from its native deposits [56]. The world's only elementary sulfur borehole mine, Osiek, is a part of the Azoty Group with the main production being agricultural fertilizers. According to the data, the Azoty Group had a comparable to the previous year financial result due to the stability of the key fertilizer segment, with the main production being agricultural fertilizers. Only in the aforementioned Osiek mine did the extraction of elemental sulfur decrease by 40% [57]. In this case, sulfur exports outside Poland amounted to 40–60% of production. These are not problems that would cause the mine to close. In Poland, the largest producer of sulfuric acid based on elemental sulfur is the Azoty Group's chemical plants in Police and Tarnów. The sulfuric acid production levels were maintained in these plants. However, the new mine Basznia II, which started mining only in 2019, had to stop its production in May 2020 (Figure 4, Table 3).

### 3.2.3. Serbia

The mineral industry has been, and it still is, an important part of the economy in Serbia since ancient times. Currently, in Serbia, there are about 200 operating mines, the most important being metallic mineral resources such as non-ferrous metals and precious

metals, and coal mining for generation of electric power [58]. Over the past decades, there has been a significant decrease in mining production of industrial minerals. The reasons for this are multiple, including insufficient investment in new technologies for exploitation, preparation and processing of industrial minerals, insufficient exploration of new reserves due to the intensive exploitation of previously defined ones, and the closure of numerous industrial plants and other facilities related to extractive industry [59]. However, the aggregate mineral industry has been modernized as a result of major investments in road construction (several highways etc.) and civil engineering [60].

The Covid-19 pandemic has significantly reduced the inflow of new investments in geological research in the Republic of Serbia in 2020 and 2021. The realization of the funds budgeted until then were realized according to plan, with certain delays caused by the declaration of a state of emergency (lockdown). The Euro Lithium Balkan company has suspended exploration in a polysource project in Valjevo-Mionica basin (Table 3).

The most important mining facilities in Serbia are Zijin Bor copper mines near the towns of Bor and Majdanpek, the Rudnik polymetallic mine, Grot and Veliki Majdan Pb-Zn mines, and Kolubara, Kostolac coal mines (Table 3). There are several large aggregate producers as well as several large producers of common clay for construction industry. An important good impact on the exploration and production of both metallic mineral resources and industrial minerals and rocks was the growing interest of foreign companies, which means that Serbia is now one of the countries with most new exploration projects in Europe. About 35 companies currently have 130 projects in the mining sector in the country, with the assessment that Serbia is an attractive destination for mining companies [61].

In the Zijin Bor Company, the total number of employees has increased for more than 10% compared to 2018. In the first six months of 2020 in Zijin Bor copper mines, despite Covid-19, the exploitation of overburden increased by 97%, wet ore increased by more than 16%, and cathode copper from its own concentrate increased by more than 18%. The impact of Covid-19 on the total economy of the company is estimated to be very low.

The impact of Covid-19 in case of the polymetallic Rudnik mine can be summarized as:

- Production—predicted production in 2020 will be at the same level as in 2019;
- Ore grade—ore grade in 2020 will be approximately the same as in previous years;
- The revenues in the period April–June decreased by 25%.

The biggest issues related to Covid-19 were during the lockdown (the state of emergency), with large organizational problems connected to less working staff (as some of the miners are from North Macedonia and had to go home during lockdown) or staff who had to stay at home with their children. The problem with approximately 22% less working staff was bypassed by exploitation from financially less favourable ore bodies, but it was easier from the mining point of view. The transport of necessary equipment and goods, as well as the export of ore, was highly impacting the production. The state program of supporting companies with a certain amount of money for salaries and taxes saved the Rudnik mine and flotation from stopping the operations (Table 3).

In the Zn-Pb mines:

- Production in this year will increase compared to 2019 but will be a few percent less than planned for 2020 due to Covid-19;
- The cost of production is higher due to unpredicted costs of Covid-19 but also due to the higher costs of concentrate processing;
- The financial effect of production will be impacted both by Covid-19 and the lower metal prices.

State support was very important for normal mining and processing operations (Table 3). No downtime or production limitations have been recorded in coal mines and power plants [62].

### 3.2.4. Sweden

Sweden is a major mining country in Europe, considering that in 2019, Sweden produced over 90% of all iron ore, and it was the largest producer of lead and zinc, and the second largest producer of silver [45,63]. The most important mining facilities in Sweden are Kirunavaara, Malmberget, Leveäniemi, and Kaunisvaara (iron ore mines) as well as Aitik (copper mine), Garpenberg, and Zinkgruvan (zinc, lead, and silver mines) and Björkdalsgruvan (gold mine). Sweden has two major mining companies with a long tradition of mining in Sweden; LKAB (the state-owned iron ore producer) and Boliden AB. Alongside these giants, there are a number of smaller mining companies active in Sweden e.g., Kaunis Iron (which started iron ore production in 2018), Zinkgruvan Mining AB, and Mandalay Resources.

During March and April 2020, Svemin—the industry organization for mines, mineral and metal producers in Sweden—performed a survey to their member companies asking questions about how the Covid-19 crisis affects their business. The main finding was that the consequences of the pandemic until that point mainly had hit companies involved in exploration activities. About 75% of all exploration companies reported that they believed that the financing situation was very problematic. It was further stressed that the insecure and inefficient permitting processes in Sweden contributed to the financing difficulties [64].

As exploration companies often are small and specialized, without income from their own mining production, they are reliant upon external capital for their activities. When the Covid-19 crisis hit, the investment climate immediately got strained, which directly affected the exploration companies. For example, a planned diamond drilling in Barsele, a gold deposit project in Västerbottens Län in Northern Sweden, was delayed by the companies Agnico Eagle Mines and Barsele (Figure 4) a few months due to Covid-19 [65]. Another exploration company, EMX Royalty, also has a number of gold-deposit projects ongoing in Sweden but did not experience any disturbance due to Covid-19 there [66]. This indicates that most exploration is continuing, despite initial disturbance due to Covid-19. Overall, if difficulties for financing exploration activities continue, it can have a long-term negative impact on the mining industry in Sweden (and Europe). An exploration permit is valid for three years in Sweden, and then it may be renewed (however, it requires special circumstances). Therefore, Svemin suggests that existing permits should be frozen—as currently, it is difficult for exploration work to continue.

The survey further found that despite the Covid-19 crisis, the Swedish mining and mineral companies have performed relatively well, and their production rates have not been particularly affected. Some companies raised that they had problems with deliveries and the relocation of key competencies (due to closed borders). The technology and service companies related to the Swedish mining industry report problems with logistics and transport, as well as declining order status and that key competences cannot be transferred. However, overall, these companies still seemed to do fairly well in March/April 2020 [64].

Production at Boliden's mines and smelters remained strong in the third quarter of 2020, despite the ongoing Covid-19 crisis. Milled production at the mines, as well as metal production at the smelters, were higher compared to the same period in 2019. The profit for Boliden's smelters, excluding revaluation of inventory, increased due to a higher volume of produced metal, higher zinc treatment charges, and higher precious metal and copper prices. In Harjavalta, a planned maintenance shutdown was carried out, which led to a slight decrease in copper and nickel production compared to the previous quarter. Precious metal production was positively affected by higher grades in input materials. Rönnskär's production of most metals decreased compared to the previous quarter. Kokkola's zinc production was on par with the previous quarter. In Odda, some disturbances in the foundry had a negative impact on zinc production [67]. In their year-end report for 2020, Boliden stress that metal prices continued to increase in quarter four due to strong demand in China, resulting in the highest coper prices for over eight years [68]. Overall, Boliden stress that their operations have adjusted well to the difficulties that the Covid-19 pandemic

has caused and that the initial offset in demand due to the pandemic has recovered in the end of the year.

LKAB states that they had a strong third quarter in 2020, with deliveries at a record high of 7.6 Mt of iron ore. However, it is argued that the effect on profit is offset by a lower dollar exchange rate, lower prices for highly upgraded iron ore products, and increased costs. The production in the third quarter decreased somewhat, which is mainly a result of measures linked to Covid-19 as well as the impact of a significant seismic event that occurred in the Kiruna mine. Taking the first three quarters of the year as a whole, both production and deliveries increased year on year [69]. So far, the main impact from Covid-19 has been related to production, as shutdowns due to maintenance were extended in order to avoid the spread of the infection, as temporary workers in the spring caused outbreaks in Gällivare/Malmberget (Table 3). LKAB further stress that this situation calls for continued flexibility and potential further measures in the operations. Some delivery volumes had to be redirected to new markets, as consumers in Europe decreased their production. It is noted in their year-end report that an increased steel production in China in the end of 2020 led to increases in iron ore prices [70].

It is also of interest to analyse the performance of the smaller mining companies currently active in Sweden, considering that their situation is somewhat different compared to the two giants. In 2020, Kaunis Iron (which started iron ore production in 2018 after buying Northland Resources facilities after their bankruptcy) reached a record year of production, despite the ongoing Covid-19 pandemic. The positive result is according to their financial presentation from 2020 much due to the increase in iron ore prices, which stems from the continued strong demand from China in combination with supply difficulties for suppliers in South America [71]. Mandalay Resources operates a gold mine in Björkdal, and they also present that their production had not been affected significantly by Covid-19. The production at the mine in 2020 was somewhat lower compared to the previous year, which is explained by production improvement projects that were installed in 2020 [72]. Lundin Mining, which operates Zinkgruvan AB in Sweden, notes that the company has adapted to the situation and has implemented preventive measures to ensure the safety of its workforce, local communities, and important stakeholders. However, production disruptions have been minimal, and no significant supply disruptions have either been the result of Covid-19 [73].

## 4. Discussion

The Covid-19 pandemic is a global factor that is changing world markets, and it is highly likely that is also affects the mining sector. For sure is that some industry players in mining will be more affected than others by the new set of challenges facing the industry. The extent as well as variation of the effects is still unknown. It seems to have hit mining operations on the verge of profitability the most, but it has affected the works that are not profitable at the moment also, such as exploration projects and closed objects (usually still extracting minerals). Projects undergoing procedures that precede mining are usually time-consuming and bureaucratic [74]. Theoretically, the best conditions are for multi-raw material (e.g., polymetallic) facilities or those whose extraction is guaranteed to be accepted, e.g., a brown coal mine and its power station.

There are no data on the stopped mining projects at the **(1) Exploration and discovery** stage. However, the survey conducted by Svemin in Sweden as well as the solution proposed by the Swedish and Polish governments have suggested that exploration permits that have been affected by the current crisis can be extended for one or more years [64,75].

The delays in exploration work for the gold Barsele project (Sweden) are mainly due to a shortage of specialists who, if they are held back by Covid-19, cannot be easily replaced.

Due to Covid-19, the possibility of the arrival of cooperants from abroad who were supposed to be involved in the process of geological research in Serbia was drastically reduced, which also caused delays in realisation of proposed investments.

Companies that conduct the **(2) Feasibility and development** stage of mining projects willingly provide data about their success. It is much more difficult for them to show difficulties and even failures. Projects are listed on stock exchanges, and misconception is quickly reflected in the share price (Figure 5).

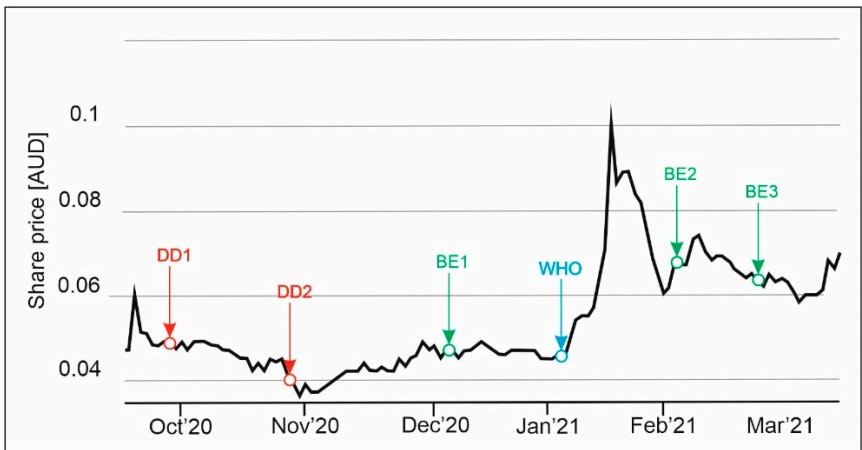

**Figure 5.** Listing of European Lithium Limited on the ASX market in the period from August 15 (2020) to March 15 (2021) (AUD—Australian Dollar) [76]. DD1—delays at the lithium production pilot plant in Hirschau, DD2—delays in finding a drilling company, BE1—published of EU Strategic Research Agenda For Batteries [77], WHO—The first vaccine to receive an emergency use validation from WHO for efficacy against Covid-19, BE2—Benchmark Mineral Intelligence published prognose of lithium price in China [78], BE3—Benchmark's Lithium ion Battery Database published prognose for Europe [79].

It is particularly important from the point of view of the European Lithium market that the Wolfsberg (Austria) exploration project seems to have overcome the difficulties and started drilling the deposit with delays. The tightening of resection and especially the closing of borders may unfortunately bring about further delays as the pilot metallurgical plant project is located in Hirschau, Germany. The company forecasts that if Covid-19 restrictions are not obstructed, it will start production in 2023 (Table 3) [46]. Wolfsberg ore contains 1% of $Li_2O$ and certainly some beryllium. These are critical raw materials for the EU economy [25]. None of the others companies reported an impact on exploration; however, three companies fear that Covid-19 will have a long-lasting impact on their business, with two others saying that it is too early to say at this stage. Long-term impacts stated include a long-lasting reduction of price, a reduction of investment, and a permanent loss of market share to non-European competitors.

However, it can be indicated for mines that are in development and at the beginning of production phase that they are still under threat and vulnerable to unforeseen changes in the raw material recipient market e.g., Basznia II mine in Poland, Cerovo, and Čukaru Peki mine in Serbia (Table 4).

The case studies analysed present mainly the **(3) Production** stage of the mining life cycle. The most important Polish mining producer, KGHM Polska Miedź, introduced the necessary restrictions related to Covid-19. Additionally, it has strongly motivated its human resources, and it can be assessed that it is managing to go through the crisis without reducing production (Figure 6). This is favoured by the conditions on the financial markets and the prices of copper and silver.

**Table 4.** Potential Covid-19 impact on stages of deposit life cycle in analysed cases.

| Stage | Impact of Covid-19 in Mining in: | | | |
|---|---|---|---|---|
| | **Austria** | **Poland** | **Serbia** | **Sweden** |
| Exploration and discovery | Low (long term) | Medium (long term) | Medium (long term) | Medium (long term) |
| Feasibility and development | Medium (short term, long term) | High (short term, long term) | Low (short term) | Medium (short term, long term) |
| Production | Medium (short term) | Medium (short term) | Low (short term) | Low (short term, long term) |
| Mine closure | Low (short term) | High (short term) | Low (short term) | Low (short term, long term) |

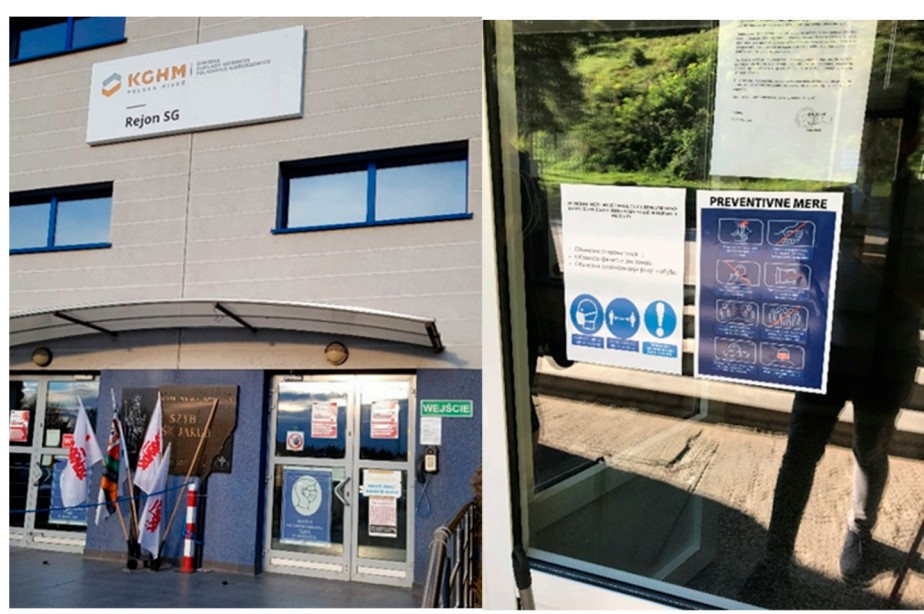

**Figure 6.** Entering one of the: (**left**) KGHM mine shafts in Poland (Fot. P. Panajew), (**right**) Rudnik mine shafts in Serbia (Fot. M. Mladenović); information on the door reminds you to keep your distance and use masks.

The greatest problems occurred in Upper Silesia (the most industrialized part of Poland), which resulted from the location of many mines (hard coal and coking coal) in a relatively small area. The number of infected miners stopped mining in one and then in groups of mines. Hard coal mines had the greatest problems with the limited availability of workforce. Over time, they also occurred in coking coal mines. Most problems were related to the first wave of the pandemic in May–June 2020. Periodic limitations of production and declines in coking coal prices on global markets translated into lower company's profits mainly in the second and third quarter of the year. Currently, sanitary regime measures have limited the spread of infection in the company. In the opinion of the JSW Management Board, the group is ready to perform its current sales contracts, including the delivery of coal to coking plants owned by the group, since the current production level and coal inventories allow for the ongoing performance of such contracts. By the same token, the production level and inventories of coke are sufficient to perform the contracts in the coke segment. However, return to restrictions associated with the next wave of Covid-19 poses a great risk to the JSW Group's operations. The increasing number of infections may lead to further restrictions influencing the demand for the JSW Group's products

(Table 4). Limitation of rail or marine transport may be a strong risk factor that may disrupt or prevent the group's sales activity.

According to today's forecasts, the continuation of performed investments is not threatened; it may only be slightly delayed because of the general limitations in the access to services and materials. Taking the Osiek sulfur mine as an example, it can be seen that reducing production causes a loss, in particular, of foreign markets, which will aggravate the problems in the future.

Generally, mining in Poland has not suffered large losses so far. All raw materials analysed for Poland are classified as key and strategic for the domestic economy [80]. The above disruptions in the activities of selected mining companies in Poland did not affect the level of raw material security in the country. The periodic limitation of production was supplemented in the following quarters of the year.

In the first six months of this year, mining in Serbia had a production growth of 3.8%. Above all, a special contribution is the exploitation of metals with a growth of more than 20%, and in this result, Zijin Bor gave a key contribution (Table 3). In August of this year, former-Minister of Mining and Energy stated that there were almost 29,000 employees in the mining sector in Serbia, which is 11.9% more than in the same period last year [81].

A survey of mining, mineral, and metal-producing companies in Sweden, conducted in March/April 2020, found that it was mainly smaller companies focusing on exploration activities that have been severely affected by Covid-19 in Sweden. To mitigate this situation, the Swedish government has suggested that exploration permits, affected by the current crisis, can be extended for one year. Local outbreaks of Covid-19 infections in Malmberget at the beginning of June 2020 affected the mining operation for LKAB. They had to temporarily shut down one facility due to the lack of workers. However, these effects were only temporary as both LKAB and Boliden, the two major mining companies in Sweden, report strong production during the third quarter of 2020, as do the active smaller mining companies. Overall, so far, the impact on the mining and mineral industry in Sweden has been relatively modest.

The Pomorzany-Olkusz mine in Poland presents in the last **(4) Closure** stage of the life cycle of mining projects. The Covid-19 crisis resulted in the acceleration of the closure of the last Zn Pb mine in Poland. However, the company's authorities ensure the continuity of metal production based on imported raw materials. The hard coal mines (Pokój, Brzeszcze) that closed in Upper Silesia are actually close to exhaustion in terms of resources, which generates higher extraction costs (resources located at great depth, on the periphery of the deposit).

## 5. Conclusions

While the outbreak has been a truly disruptive event, the mining and metals sector has dealt with its impact extremely well, leading an effective response. Initially, the impact is on day-to-day operations and ways of working. As well as measures such as social distancing, customers close access to their sites and try to maintain operations with minimal staff. The clear majority of customers are trying to carry on with business as usual. As a result, many mines have remained operational and productive during the pandemic, despite having less people on site. However, business continuity has come at a cost due to the added expenses of new processes, procedures, protocols, health testing equipment, and support for the workforce. The decline in commodity prices that was seen at the beginning of the pandemic has been completely offset in recent months.

It must be clearly emphasized that most of the factors described above constituted obstacles at different stages of the mining project life cycle also before the Covid-19 pandemic. The Covid-19 crisis has in some cases made easy to medium obstacles become insurmountable for exploration and mining companies.

Each of the analysed countries has mineral resources and active mines that affect the state of its economy. The deposit management policy is carried out in various ways, but

everywhere, new facilities are searched for, mining is carried out in active plants, and closed and reclaimed mines are closed.

*5.1. Long-Term Impact of Covid-19*

The most difficult situation seems to be for companies that specialize only in the prospecting and exploration of deposits. This is due to the fact that they relate to the distant future and due to the fact that companies are not willing to provide data because in case of problems with project implementation, they will be negatively perceived. As described in the article's first part, these types of projects are associated with a long implementation period and require relatively large financial outlays without a guarantee of success. Therefore, difficulties in obtaining financing and, consequently, in the implementation of new projects at an early stage can be expected. Political instability strengthens the financial risk of such a product and weakens the position of the seller. Such a situation will mean that the effects of the current situation with Covid-19 will be felt only in a few years, when there may be a reduction in new projects ready for implementation. As well, often they do not have enough staff to quickly replace those who are stopped by the Covid-19 restrictions. This effect is enhanced by the closing of state borders, which makes it difficult to search for replacements. The Swedish government has offered to extend the validity of the exploration permit. In Poland, after the state of pandemic emergency was announced, the expiry of the concession period was suspended until further notice, creating losses related to stopping the search for new deposits that are difficult to assess. Stopping exploration works delays the discovery of potential objects for exploitation. As a result, this delay may result in other developments of the area above the deposit [82], which can make it impossible to extract the mineral resources in the future or significantly increase mining costs.

Assistance in the exploitation phase is easier to justify, as the lack of extraction may mean a lack of raw materials for the production of important goods, electricity, etc. In Poland, mining companies benefited from subsidies when they had to reduce their working hours due to infection of the crew with virus. Hard coal mines benefited greatly from this. Visible problems included mainly staffing in mines and limited problems with selling products.

*5.2. Short-Term Impact of Covid-19*

Numerous strategic uses of lithium for portable electronic devices, stationary energy storage and electric vehicles (EVs) make it one of the few mineral commodities in the world that is welcomed with open arms. Therefore, the delay in opening the Wolfsberg (Austria) mine with an expected production of 10,129 t/pa lithium should be assessed as a serious loss.

The short-term effects of Covid-19 crisis mainly include the suspension or limitation of exploitation/production resulting mainly from the issue of employee absenteeism caused by quarantine or pandemic restrictions. In contrast, for projects located in other European countries, the suspension of production due to epidemiological reasons was clearly felt. In Austria, the talc mines stopped production for 2 weeks. In Ireland, the lockdown stopped the production of two large mines of zinc Boliden Tara Mines DAC [83].

Another typical short-term effect of Covid-19 is shortening the lifetime of mining projects in the closure phase. However, the earlier-than-expected mine closure does not generally affect the industry, as their owners were prepared for such a scenario in the event of a sharp downturn in the markets. Possible shortening or extending the life of projects in the declining phase by 1–2 years does not affect the condition of the industry as a whole. Slowing down reclamation causes the negative environmental impact of the closed mine to be maintained longer than expected. The costs are borne by local communities and the environment.

It must be admitted that this is a first forecast of the impact caused by Covid-19 in the mining sector. Collecting the data so early on was difficult, as many companies have not yet summarized the losses and profits from this period and are still uncertain about

possible additional future impacts of the pandemic. Several companies were so surprised by the Covid-19 pandemic that they suddenly shut down without leaving any information behind as to why. However, the authors decided to publish conclusions obtained on the basis of preliminary and incomplete data. This is a moment for warning mining companies and resource management as well as securing supplies of mineral resources. The results indicate that the prolongation of the Covid-19 crisis carries a real risk of permanently disrupting the supply chain of raw materials.

**Author Contributions:** Conceptualization: A.G., H.C., A.K.-N.; methodology: A.G., H.C.; software: A.G., A.K.-N., S.G.; validation: A.G., A.K.-N., H.C.; formal analysis: A.G., A.K.-N., H.C., M.T., L.W., V.S.; investigation: A.G., A.K.-N., H.C., M.T., L.W., V.S.; resources: A.G., A.K.-N.; data curation: A.G., A.K.-N.; writing—original draft preparation: A.G., A.K.-N., M.T., L.W., V.S., H.C.; writing—review and editing: A.G., A.K.-N., M.T., L.W., V.S., S.G.; visualisation: A.G., S.G., A.K.-N.; supervision: A.G., A.K.-N.; project administration: A.G.; A.K.-N.; funding acquisition: A.K.-N., A.G. All authors have read and agreed to the published version of the manuscript.

**Funding:** This article has been supported by the Polish National Agency for Academic Exchange under Grant No PPI/APM/2019/1/00079/U/001.

**Data Availability Statement:** The study did not involve humans or animals.

**Acknowledgments:** The authors thank three anonymous reviewers for their constructive commentary on a previous draft as well as Krzysztof Galos for constructive comments on the draft version of this manuscript. We would like to thank Marko Mladenović and Paweł Panajew for taking the photos for Figure 6.

**Conflicts of Interest:** The authors declare no conflict of interest.

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
