# Peer review of "Impact of Covid-19 on the Mining Sector and Raw Materials Security in Selected European Countries"

_resources, doi:10.3390/resources10050039_

Round 1

Reviewer 1 Report

This paper is interesting but needs several writing editing. It is important a comparison between program before this event and differences after this epidemic event. However, please describe about ore deposit descriptive model of lithium deposits in this paper.

Reviewer 2 Report

This is a good empirical study on the impact of Covid-19 on the mining sector and raw materials security in several European countries. I've found great merits of this paper on its findings (interpretation aspects), but there are still some reworks needed to be improved "

1) The introduction should cleraly state on which literature this paper brings new added value or what it brings new to current literature;

2) I would like to see a literature review section dealing with the two topics analysed in this study: Covid-19 impact on mining; and the mining industry/extraction literature. Authors can present some Covid-19 cases in the world related to broader labour issues(eg. 10.1177/0896920520929966; 10.1080/15387216.2020.1780929; 10.1016/j.worlddev.2020.105315) and also to read and cite many books and papers in extractive literature, mainly dealing with mining both before and during the pandemic (eg. see https://www.ceeol.com/search/article-detail?id=531737; or see mining in different European contexts : see Vesalon L.'s articles on gold mining at Rosia Montana etc, see Szczepanska's articles on Poland mines etc); besides these suggestions, authors cited many academic works (the reference list is large) and most of those works can also be used to frame the literature review of the paper;

3) The findings are good, but the conclusions should be expanded by shortly presenting the limitations of the study and the follow-up research (eg. how other researchers could develop their findings). 

Reviewer 3 Report

The authors have conducted a survey to assess the short- and long-term impacts of Covid-19 pandemic on the mining industry, using a selected number of European countries as case studies. 

The research topic is interesting and definitely falls within the scope of "Resources". However, there are some points that need to be clarified and improved before the paper becomes acceptable for publication. 

First, there are many unnecessary repetitions in the text. For example (these are not the only cases):

  • Lines 53-54: "..., it lowered the demand for and sales of products and stopped many economic activities..." and lines 58-59 "...it caused a drastic decrease in the demand for basic goods and services..."
  • Lines 146-147: "...In order to assess the impact of Covid-19 on the mining sector, its effects should be taken into account at a given stage of mining project life-cycle...." and lines 161-162: "...Considering the foregoing, in order to assess the impact of Covid-19 on the mining 161 sector, its effects should be taken into account at a given stage of the mining project development cycle...".

Second, the "Materials and methods" section needs to be considerably improved. For example, in lines 125-126 it is written that "...The work is based mainly on the analysis of statistical data in databases in the four countries". You should mention the databases and provide the related links. Further, in lines 127-128 it is referred that "...The authors conducted surveys in selected mining centres and with mining companies (Table 2). Much information was obtained from the official reports of mines and companies....". Yet, Table 2 provides details just for one mine. Also, there are no details regarding the surveys (e.g. were they conducted by means of questionnaires? If yes, further details should be given, such as the construction of the questionnaire, the sample used, the response rate, etc.). This is an important weakness of the paper, because it jeopardises the reliability, reproducibility and understanding of the results (i.e. it's hard to understand if they come from the databases or the communication with the companies). Finally, to my opinion, Section 3.1 should be embedded in Section 2 because it does not present research results; it describes the potential impacts of Covid-19 on different stages of the mining activities.

Finally, there are several grammar and spelling mistakes and both American and British English are used. 

A minor comment: in line 606 you refer "Table 5". Do you mean "Figure 5"?

Round 2

Reviewer 2 Report

Authors have done a nice revision. The paper is much improved. I would like to accept this paper for publication. 

Author Response

We have added one more literature item suggested by Reviewer 2

Reviewer 3 Report

The authors have amended almost all the comments of the first review. I suggest however to delete Table 2 and add general (and anonymous) information regarding the mining companies that participated in the survey (e.g. how many companies participated in the survey, from which countries, what kind of mineral deposits they own, etc.).
